# CrossTVR: Multi-Grained Re-Ranker for Text Video Retrieval with Frozen Image Encoders

## Abstract

State-of-the-art text-video retrieval (TVR) methods typically utilize CLIP and cosine similarity for efficient retrieval. Meanwhile, cross attention methods, which employ a transformer decoder to compute attention between each text query and all frames in a video, offer a more comprehensive interaction between text and videos. However, these methods lack important fine-grained information as they directly compute attention between text and video-level tokens for computation cost consideration. To address this issue, we propose a fine grained re-ranker named CrossTVR with a novel multi-grained video text cross attention module to capture fine-grained multimodal information from both frame and video level. The re-ranker only improves the top K similar results from the existing cosine similarity network, so the inference can be performed efficiently. To reduce the additional training cost by the multi-grained cross attention module, in the second stage, we freeze the vision backbone and only train the video text cross attention matching header in the second stage, enabling scalability to larger pre-trained vision models like ViT-G, resulting in improved retrieval performance. Experiments on text video retrieval datasets demonstrate the effectiveness and scalability of our proposed CrossTVR compared to state-of-the-art approaches.

## 1 Introduction

Text Video Retrieval (TVR) is a pivotal task in the domain of visual-language understanding, with the objective of retrieving pertinent videos given a natural language text and vice versa. Presently, there exist three primary approaches to tackle the challenge of TVR. The first approach Luo et al. (2021); Liu et al. (2022); Wang et al. (2022b); Fang et al. (2021); Ma et al. (2022) involves the independent mapping of text and videos to a shared embedding space utilizing dual encoders. Subsequently, a lightweight cosine similarity calculation is performed to enable approximate nearest neighbor search. Although this approach is efficient, its accuracy is constrained due to the straightforward dot product-based interaction between vision and text within the shared embedding space. The second approach Gorti et al. (2022); Luo et al. (2021) employs cross-attention transformers to establish a comparison between each word and the frames in the video, thereby enabling a comprehensive interaction between text and videos. However, this approach incurs high computational costs during inference, thereby limiting its potential to further explore fine-grained interactions at the video entity and event level. The third approach, as proposed by Miech et al. (2021); Li et al. (2021a); Wang et al. (2022a), combines elements of both aforementioned types. It begins by eliminating videos that lack commonalities with the text description, utilizing a similarity-based approach, and subsequently employs cross-attention to obtain the final set of promising candidates. Meanwhile, the recent advancements in multimodal contrastive learning model CLIP Radford et al. (2021) have facilitated the attainment of state-of-the-art performance by integrating a similarity-based approach with the CLIP paradigm Luo et al. (2021); Liu et al. (2022); Wang et al. (2022b); Gorti et al. (2022); Fang et al. (2021); Ma et al. (2022).

In this research paper, we aim to investigate the third approach based on the CLIP paradigm and propose a re-ranker called CrossTVR. The primary objective of this re-ranker is to further explore the comprehensive and fine-grained interaction between text and vision tokens at both the frame and video levels. To illustrate the process of generating visual tokens, as shown in Figure 1, we describe the procedure of generating $N$ visual tokens for each of the $T$ frames in a given video, resulting in a total of $T \times N$ vision tokens. However, directly concatenating all the visual tokens

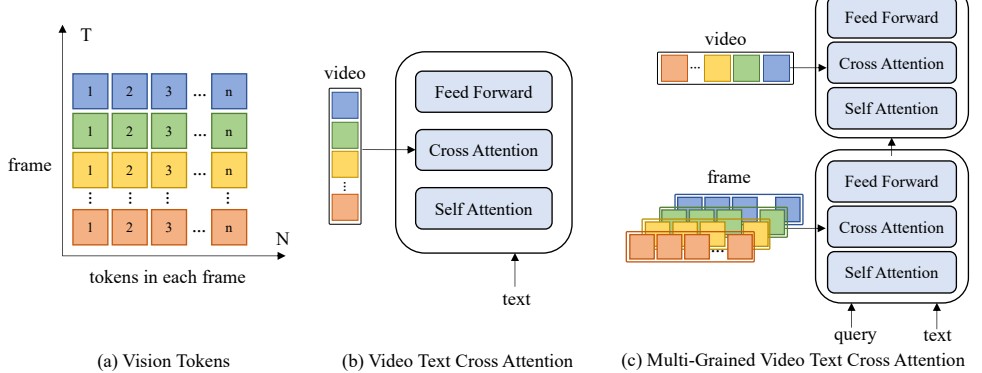

(a) Vision Tokens  (b) Video Text Cross Attention  (c) Multi-Grained Video Text Cross Attention

Figure 1: Figure (a) illustrates the encoding process of a video with $T$ frames. The vision encoder generates $T \times N$ tokens, with each frame containing $N$ tokens. Tokens belonging to the same frame are represented by the same color. Existing cross attention methods (b) Gorti et al. (2022); Luo et al. (2021); Miech et al. (2021) solely compute video-level attention. In our proposed approach (c), we introduce multi-grained cross attention that operates at both the frame level and the video level.

of each frame for cross-attention can be resource-intensive in terms of memory and time. Consequently, existing cross-attention-based techniques Gorti et al. (2022); Luo et al. (2021); Miech et al. (2021) compute cross-attention between text and video tokens, which are generated either through mean pooling or by selecting only the [CLS] token in each frame. Regrettably, this approach may result in the loss of comprehensive and fine-grained spatial and temporal information, such as subtle movements and small objects at the entity and event levels. To overcome this limitation, we propose a multi-grained video-text cross-attention module that separately computes frame-level spatial text attention and video-level temporal text attention. Specifically, the visual tokens derived from the pre-trained image vision encoder for each frame are well-suited for spatial modeling, such as small entities. By computing spatial frame-text cross-attention between text tokens and $N$ visual tokens of each frame, we can extract spatial-enhanced multimodal features. Furthermore, to capture the temporal video-text interaction, such as subtle movement, we compute cross-attention between the multimodal features and representative video tokens (selected by a token selector Liu et al. (2022)) to capture salient temporal semantic features.

Recent advancements in Contrastive Language-Image Pre-training (CLIP) Radford et al. (2021) have paved the way for improved video representation learning. By leveraging a frozen vision encoder Lin et al. (2022) and a learnable lightweight transformer decoder, high-quality video representations can now be learned without the need for end-to-end fine-tuning. In our proposed multi-grained video text cross attention module, we employ a sequence of learnable query tokens to extract relevant visual features from the vision-language pre-trained model. This allows us to freeze the vision encoder and only train the cross attention module, resulting in significant reductions in computation and memory resources. As a result, we can leverage very large image architectures, such as the ViT-G Sun et al. (2023) vision encoder, which has been open-sourced. Building upon these ideas, we propose a two-stage text-video retrieval architecture. In the first stage, we employ a standard cosine similarity network. In the second stage, we introduce the multi-grained video text cross attention module, which combines fine-grained visual and text tokens at the frame and video levels, respectively. By incorporating a hard mining strategy, this module can capture subtle differences among well-selected candidates, leading to improved re-ranking results.

Our proposed CrossTVR is extensively evaluated together with representative cosine similarity methods on various text-video retrieval benchmarks, including MSRVTT Xu et al. (2016), VA-TEX Wang et al. (2019), LSMDC Rohrbach et al. (2017), ActivityNet Fabian Caba Heilbron & Niebles (2015), and DiDeMo Anne Hendricks et al. (2017). Through these evaluations, we demonstrate that our approach obviously enhances the performance of these methods, achieving state-of-the-art results across all benchmarks. In summary, The main contributions of this work are as follows:

- We propose a multi-grained re-ranker called CrossTVR, which achieves comprehensive interaction between text and video at the frame level and video level.

- As a re-ranker, our method can be widely applied to existing cosine similarity-based methods and effectively improve the SOTA retrieval performance with marginal additional computation cost.

- Benefiting from our freezing visual coder training method, our approach can scale to larger pretrain visual models with small computational resources.

## 2 RELATED WORK

**Text-Video Retrieval**  Various approaches have been proposed to address text-video retrieval tasks, which can be broadly categorized into three types: cosine similarity-based, cross attention-based, and hybrid methods combining both approaches. Cosine similarity-based approaches Lei et al. (2021); Bain et al. (2021); Luo et al. (2021); Fang et al. (2021); Cheng et al. (2021) typically leverage contrastive learning from CLIP Radford et al. (2021) to compute the cosine similarity between video and text embeddings. CLIP4Clip Luo et al. (2021), CLIP2Video Fang et al. (2021), and TS2Net Liu et al. (2022) explore different mechanisms for transferring knowledge from pre-trained CLIP to video retrieval tasks. HBI Jin et al. (2023) combines cosine similarities at entity, action, and event levels using a weighted sum. However, the accuracy of cosine similarity-based methods is limited due to the simplicity of the vision-text interaction model defined by the dot product in the joint embedding space. Cross attention-based methods, on the other hand, employ attention modules to enable fine-grained multimodal interaction. X-Pool Gorti et al. (2022) introduces a scaled dot product-based cross-modal attention model that generates an aggregated video representation based on the text's attention weights over the frames. CLIP4Clip-tightTransf Luo et al. (2021) utilizes a cross attention transformer to capture the complex relationship between arbitrary-length text and videos. However, the cross attention transformer suffers from optimization difficulties and high training costs due to its computational complexity, which limits its potential to explore the multi-grained correspondence between text and videos. In order to leverage the benefits of different approaches, certain methods Miech et al. (2021); Li et al. (2021a); Wang et al. (2022a) have adopted a coarse-to-fine strategy, wherein a fast cosine similarity measure is initially employed to obtain coarse retrieval candidates, followed by the utilization of cross attention to obtain the desired final results. In this work, we follow this approach and propose a novel multi-grained video text cross attention module, which effectively captures comprehensive and fine-grained interactions at both the frame-level and the video-level.

**Vision-Language Pre-training**  Vision-language pre-trained models have demonstrated promising outcomes in various image-language tasks, including image retrieval, visual question answering, and image captioning. These models typically employ a shared self-attention transformer encoder that accommodates multi-modal inputs Li et al. (2020a); Su et al. (2020); Zhou et al. (2020). Additionally, they utilize cross attention transformers to fuse different modalities Lei et al. (2021); Li et al. (2021b); Wang et al. (2022c) or employ contrastive loss to align text and visual embeddings Bain et al. (2021); Jia et al. (2021); Radford et al. (2021). Similarly, video-language pre-trained models Li et al. (2021a); Bain et al. (2021); Li et al. (2020b); Luo et al. (2020); Wang et al. (2022a); Zeng et al. (2022) have been proposed specifically for video-language tasks. Meanwhile, InternVideo Wang et al. (2022d) and CLIP-ViP Xue et al. (2022) initialize the model from CLIP and further pre-train it on video datasets. However, these methods necessitate additional training on large-scale video data and cannot utilize off-the-shelf vision-language pre-trained models. Following recent works Luo et al. (2021); Liu et al. (2022), instead of pre-training, our method aims to transfer the knowledge from the pre-trained model to video-text retrieval task. Thanks to the frozen vision encoder strategy, we can use very large vision-language pre-trained models to further boost up the performance.

**Fine Grained Attention**  Fine-grained attention mechanisms have been widely employed in various computer vision tasks. For instance, in fine-grained visual categorization, where the goal is to distinguish categories with subtle differences, recent methods such as TransFGAT He et al. (2021), FeatureFusion Wang et al. (2021), and DualCL Zhu et al. (2022) utilize fine-grained self-attention and cross-attention mechanisms to fuse vision features from multiple levels. In the context of vision-language pre-training, FLIP Cho et al. (2022) models fine-grained semantic alignment by considering token-wise similarity between visual and textual tokens. Similarly, ABLEF Li et al. (2021b) and BLIP Li et al. (2022) employ fine-grained cross-attention mechanisms to compute interactions between text and image tokens. In our work, we extend these cross-attention mechanisms from the image level to the video level. In the domain of video retrieval, TS2Net Liu et al. (2022) aims to model fine-grained spatial and temporal information of input video samples. To eliminate background tokens that may contain irrelevant information and dominate the final video representation, TS2Net selects the most informative tokens from each frame as frame-wise video embeddings and calculates the similarity between the text and these selected tokens. On the other hand, DRL Wang et al. (2022b) introduces a weighted token-wise interaction method to facilitate comprehensive inter-

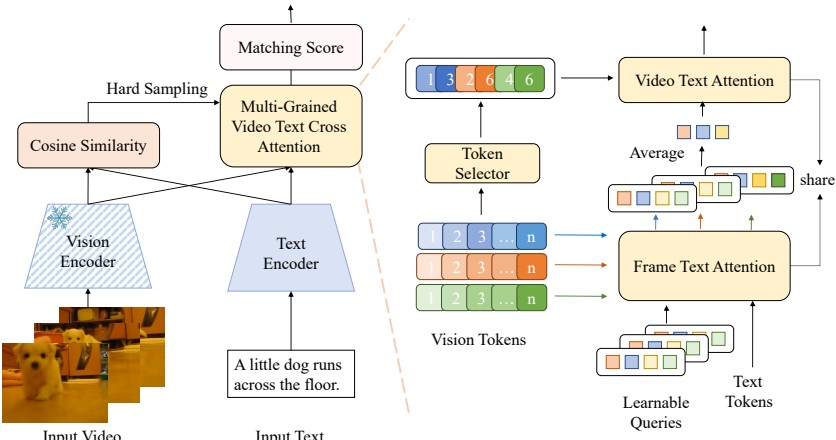

Figure 2: The training pipeline of CrossTVR using a video of 3 frames as an illustrative example. Vision tokens of same color correspond to the same frame. Throughout the training process, the vision encoder is frozen. Initially, the cosine similarity header is employed to identify challenging negative samples, which are subsequently subjected to the calculation of fine-grained cross attention matching scores. The multi-grained video text cross attention mechanism is then employed to compute spatial text attention at the frame level and temporal text attention at the video level. Finally, a single fully connected layer is utilized to generate the matching score.

actions between sentence tokens and video frame tokens. In contrast, our method computes multi-grained cross-attention between the text and all tokens of each frame, thereby further enhancing the performance of comprehensive and fine-grained video-text interaction.

## 3 METHOD

The primary objective of text-video retrieval (TVR) is to identify the most relevant videos based on a given text query. To achieve this, our TVR framework consists of two training stages. In the initial stage, we train a cosine similarity-based network (e.g. TS2Net Liu et al. (2022) with ViT-B/32) using contrastive learning. This stage focuses on efficient retrieval by employing cosine similarity-based methods. In the subsequent stage, we apply the re-ranker called CrossTVR to be described in Section 3.1, which incorporates multi-grained cross attention mechanisms for more detailed matching. Figure 2 provides the detailed training pipeline of the second stage, which includes a frozen vision encoder, a text encoder, and a multi-grained video text cross attention module.

### 3.1 MULTI-GRAINED VIDEO TEXT CROSS ATTENTION

In this section, we propose an architecture for multi-grained video-text cross attention that effectively incorporates both frame-level and video-level information. Existing multimodal Transformer-based methods for Text-to-Video Retrieval (TVR) Feichtenhofer et al. (2019); Luo et al. (2021); Xue et al. (2022) often compute attention directly between text and video tokens, which results in a loss of fine-grained visual appearance representations. To address this issue, we introduce separate modules for frame-text attention and video-text attention. These modules are depicted in Figure 1. The frame-text attention module captures the fine-grained frame-text interaction by computing attention between the given text and each frame, while the video-text attention module aims to learn the global video-text multimodal representation which aggregating spatial interaction information of all frames. This design choice allows for more efficient parameter sharing between the two modules, facilitating training with fewer parameters. Furthermore, the multi-grained but parameter-sharing architecture enhances multimodal representation learning and enables spatial and temporal attention to mutually benefit each other.

**Frame Text Attention** We achieve frame-level vision-language fusion using a cross attention module. The module concatenates a fixed number of learnable query embeddings and text embeddings as query input to the cross attention transformer. The queries interact through self-attention layers and then with each frame feature of the video through cross attention individually. This approach enables queries to retrieve both the overall information (represented by the [CLS] token) and the salient spatial token in each frame. This helps to maintain the fine-grained spatial features of the video as much as possible, thereby facilitating their fusion with textual information. As a result, we

obtain the frame-level spatial enhanced queries for the video using the following equations:

$$\mathbf{z}_{t,1} = Attn(Concat(Q, X), V_t) \tag{1}$$

$$\mathbf{z}_{t,l} = Attn(\mathbf{z}_{t,l-1}, V_t), \ l = 2, ..., L \tag{2}$$

$$X_{frame}(t) = Avg(\mathbf{z}_{t,L}^1, \mathbf{z}_{t,L}^2, \cdots, \mathbf{z}_{t,L}^{N_Q}) \tag{3}$$

Here, $Q$ represents $N_Q$ learnable query embeddings, $X$ is text tokens, $V_t$ is the vision tokens for the $t_{th}$ frame, $\mathbf{z}_{t,l}^i$ is the $i$-th token of the $l$-th spatial cross attention layer's output given the $t$-th frame tokens, and $X_{spatial}(t)$ selects and averages the first $N$ tokens of the last cross attention layer's output $\mathbf{z}_{t,L}$. As a result, $X_{spatial}$ of size $T$ is obtained as the spatial enhanced text queries.

**Video Text Attention**  We compute the video-level interaction through video-text cross attention, using the spatial enhanced text queries as queries and video feature tokens as values. However, the way in which we aggregate multi-frame information has an important impact on video-level interaction. Using only [CLS] tokens tends to lose detailed information while introducing all tokens brings irrelevant and redundant information. To address this, we draw on the token selector module described in Liu et al. (2022) to obtain the salient objects and movement in terms of temporal. As we mentioned before, this module shares parameters with the frame text attention module for training efficiency. Given a sequence of visual tokens, the token selector module employs an MLP followed by a Softmax layer to predict the importance score for each token and select the $M$ most informative tokens for each frame. This can be expressed in the following equations:

$$V_{select}(t) = TokenSelector(V_t, M) \tag{4}$$

$$\mathbf{z}_1 = Attn(Concat(X_{frame}(t)), Concat(V_{select}(t))) \tag{5}$$

$$\mathbf{z}_l = Attn(\mathbf{z}_{l-1}, Concat(V_{select}(t))), \ l = 2, ..., L \tag{6}$$

Here, the video text module computes the cross attention between $M \times T$ flattened visual tokens (as key and values) and $T$ spatial enhanced tokens (as queries) of $\mathbf{z}_L$.

## 3.2 FROZEN IMAGE ENCODER

Recently, CLIP Radford et al. (2021) has demonstrated promising feature transferability for various downstream image tasks, such as classification Zhou et al. (2021) and segmentation Lüddecke & Ecker (2021). To extend CLIP to video tasks, EVL Lin et al. (2022) and AIM Yang et al. (2023) use frozen CLIP image encoder and a learnable lightweight decoder or a sequence of adaptors to adapt it for single modal video recognition tasks. In contrast, our cross attention header aims to transfer fine-grained vision features based on text queries for multimodal understanding. Specifically, in the second stage of CrossTVR, we select two state-of-the-art image pre-trained vision transformers as the frozen vision encoder: ViT-B/32 from CLIP Radford et al. (2021) and ViT-G/14 from EVA-CLIP Sun et al. (2023). We refer to them as CrossTVR Base and CrossTVR Large respectively, for comparison with recent TVR methods Luo et al. (2021); Liu et al. (2022). For both CrossTVR Base and CrossTVR Large, since the cross attention header can explore fine-grained relationship between video and text during re-ranking, we can use a relatively smaller vision encoder ViT-B/32 for cosine similarity to ensure efficient retrieval in the first stage. In the second stage of CrossTVR Large, since ViT-G/14 is frozen, the total number of training parameters is significantly smaller than an end-to-end finetuned ViT-G network Luo et al. (2021), reducing GPU memory usage by 91%. Notably, our multi-grained video text cross attention and frozen CLIP model strategies complement existing similarity-based approaches Luo et al. (2021); Liu et al. (2022); Gorti et al. (2022).

## 3.3 TRAINING

To train CrossTVR, we follow a two-stage approach. In the first stage, we use contrastive learning to train the cosine similarity network. In the second stage, we freeze the vision encoder and train the cross attention header via video text matching. Since CrossTVR seeks to learn fine-grained interaction between video and text representation, we adopt the hard negative mining strategy from ALBEF Li et al. (2021b) to generate informative negative pairs during training. Here, we consider a negative video-text pair to be hard if they share similar semantics but differ in fine-grained details. Specifically, for each video in a mini-batch, we sample one negative text from the same batch following the contrastive similarity distribution, where texts with higher matching scores with the

video are more likely to be sampled. Similarly, we also sample one hard negative video for each text. With these training pairs, we compute the text-video matching loss $\mathcal{L}_{\text{tvm}}$ as follows:

$$\mathcal{L}_{\text{tvm}} = \mathbb{E}_{(X,V) \sim D} \text{H}(\mathbf{y}, \mathbf{p}(X, V)) \tag{7}$$

Here, $D$ represents the set of positive and negative sample pairs, H denotes the cross-entropy loss, $\mathbf{y}$ is a 2D one-hot vector representing the ground-truth label, and $\mathbf{p}$ denotes the multi-grained video text cross attention matching score.

### 3.4 INFERENCE

During the inference process, we begin with computing a rough similarity matrix through the cosine similarity network. Subsequently, we utilize multi-grained video text cross attention to re-rank text-video pairs based on their top similarity scores. To elaborate further, suppose we have a query text and a vast video database containing $M$ videos. We initially compute $M$ similarity scores between the text and $M$ videos using cosine similarity. Then, we select the top $K$ videos (where $K \ll M$) for fine-grained re-ranking using cross attention. Finally, we determine the final rank of the top $K$ videos by adding the scores of cosine similarity and fine-grained matching. Since $K$ is considerably smaller than the whole dataset, the inference of TVR can be performed efficiently.

## 4 EXPERIMENTS

### 4.1 EXPERIMENTAL SETTINGS

**Datasets.** We conduct experiments on the following five popular text-video retrieval benchmarks: MSR-VTT Xu et al. (2016) comprises 10,000 videos, each with 20 captions and a duration of 10 to 32 seconds. We adopt the 'Training-9K' split. ActivityNet Krishna et al. (2017) comprises 20,000 videos collected from YouTube. VATEX Wang et al. (2019) comprises 34,991 video clips, each with multiple captions. LSMDC includes 118,081 videos and an equal number of titles obtained from 202 movies, with each video ranging from 2 to 30 seconds in length. DiDeMo Anne Hendricks et al. (2017) comprises more than 10,000 videos and 40,000 captions, with all videos are segmented into 5-second intervals.

**Evaluation Metrics.** To evaluate the performance of our model, we use standard retrieval metrics for both Text2Video (T2V) and Video2Text (V2T): recall at rank K (R@K, higher is better) and mean rank (MnR, lower is better). R@K calculates the percentage of test samples that find correct results within the top K query samples. We select K=1,5 in the experiment. MnR represent the mean of the ground-truth results in the retrieval ranking.

**Implementation Details.** In the first stage, we choose two representative networks TS2Net (ViT-B/32) Liu et al. (2022) and CLIP-ViP (ViT-B/32) Xue et al. (2022), for cosine similarity computation. The weights of TS2Net are initialized from publicly available CLIP checkpoints, which are pretrained only on images. The weights of CLIP-ViP are initialized from its official released checkpoint, which is pretrained on the video dataset HD-VILA-100M Zhao et al. (2022). In the second stage, we have the option to use either ViT-B or ViT-G as the visual encoder for cross attention. For ViT-B/32, we freeze the visual encoder of TS2Net (ViT-B/32) and CLIP-ViP (ViT-B/32), and reuse them for the cross attention header. For ViT-G/14, we initialize it from EVA-CLIP Sun et al. (2023) and freeze it during training. The remaining modules are randomly initialized. We set M=4 for the token selector and K=15 for the re-ranking stage. The max query text length and max video frame length are 32 and 12 for MSR-VTT, VATEX, and LSMDC. For ActivityNet and DiDeMo, the max query text length and max video frame length are set 64 and 64. The model is trained using the Adam optimizer Kingma & Ba (2014), with an initial learning rate of 1e-4 for the trainable modules. All learning rates follow a cosine schedule with a warm-up setup. Training is conducted on a single NVIDIA A100 card, with a batch size of 128 for all datasets.

### 4.2 COMPARISON WITH STATE-OF-THE-ARTS

**MSR-VTT.** Table 1 provides a comprehensive comparison between our proposed method and existing approaches on the MSRVTT dataset. With the ViT-B/32 as vision encoder, our method

| Method | Text2Video | | | Video2Text | | |
|---|---|---|---|---|---|---|
| | R@1 | R@5 | MnR | R@1 | R@5 | MnR |
| CE Liu et al. (2019) | 20.9 | 48.8 | 28.2 | 20.6 | 50.3 | 25.1 |
| MMT Gabeur et al. (2020) | 26.6 | 57.1 | 24.0 | 27.0 | 57.5 | 21.3 |
| FrozenBain et al. (2021) | 31.0 | 59.5 | - | - | - | - |
| CLIP4ClipLuo et al. (2021) | 44.5 | 71.4 | 15.3 | 42.7 | 70.9 | 11.6 |
| CenterCLIPZhao et al. (2022) | 44.2 | 71.6 | 15.1 | 42.8 | 71.7 | 10.9 |
| CAMoECheng et al. (2021) | 44.6 | 72.6 | 13.3 | 45.1 | 72.4 | 10.0 |
| CLIP2VideoFang et al. (2021) | 45.6 | 72.6 | 14.6 | 43.5 | 72.3 | 10.2 |
| CLIP2TVGao et al. (2021) | 45.6 | 71.1 | 15.0 | - | - | - |
| X-PoolGorti et al. (2022) | 46.9 | 72.8 | 14.3 | - | - | - |
| X-CLIPMa et al. (2022) | 46.1 | 73.0 | 13.2 | 46.8 | 73.3 | 9.1 |
| OmniVLWang et al. (2022a) | 47.8 | 74.2 | - | - | - | - |
| TS2-NetLiu et al. (2022) | 47.0 | 74.5 | 13.0 | 45.3 | 74.1 | 9.2 |
| HBIJin et al. (2023) | 48.6 | 74.6 | 12.0 | 46.8 | 74.3 | 8.9 |
| CLIP-ViPXue et al. (2022) | 50.1 | 74.8 | - | - | - | - |
| TS2-Net + Ours(Base) | 50.0 | 75.7 | 12.0 | 47.1 | 76.6 | 8.1 |
| TS2-Net + Ours*(Base) | 52.8 | 78.9 | 11.2 | 52.5 | 77.3 | 8.9 |
| CLIP-ViP + Ours(Base) | 51.9 | 76.8 | 14.3 | 50.6 | 77.4 | 9.4 |
| CLIP-ViP + Ours*(Base) | 56.9 | 80.0 | 11.5 | 55.3 | 78.3 | 8.5 |
| TS2-Net + Ours(Large) | 54.0 | 77.5 | 11.8 | 51.3 | 78.3 | 8.1 |
| TS2-Net + Ours*(Large) | 58.3 | 81.3 | 11.4 | 57.3 | 79.9 | 9.5 |
| CLIP-ViP + Ours(Large) | 54.2 | 77.2 | 14.2 | 53.4 | 78.7 | 9.3 |
| CLIP-ViP + Ours*(Large) | 57.3 | 79.8 | 13.7 | 57.9 | 80.2 | 9.2 |

Table 1: Performance comparison on MSR-VTT. * means DSL Cheng et al. (2021) is applied as post-processing operation during inference.

| Method | Text2Video | | | Video2Text | | |
|---|---|---|---|---|---|---|
| | R@1 | R@5 | MnR | R@1 | R@5 | MnR |
| CLIP4ClipLuo et al. (2021) | 40.5 | 72.4 | 7.5 | 41.4 | 73.7 | 6.7 |
| CenterCLIPZhao et al. (2022) | 43.9 | 74.6 | 6.7 | 44.5 | 75.7 | 6.5 |
| X-CLIPMa et al. (2022) | 44.3 | 74.1 | 7.9 | 43.9 | 73.9 | 7.6 |
| TS2-NetLiu et al. (2022) | 41.0 | 73.6 | 8.4 | - | - | - |
| HBI Jin et al. (2023) | 42.2 | 73.0 | 6.6 | 42.4 | 73.0 | 6.5 |
| CLIP-ViP Xue et al. (2022) | 51.1 | 78.4 | - | - | - | - |
| TS2-Net + Ours(Base) | 49.0 | 78.4 | 7.1 | 47.4 | 78.8 | 7.4 |
| TS2-Net + Ours*(Base) | 55.0 | 82.1 | 8.2 | 57.1 | 82.3 | 6.8 |
| CLIP-ViP + Ours(Base) | 55.0 | 81.4 | 5.3 | 53.1 | 82.2 | 5.0 |
| CLIP-ViP + Ours*(Base) | 62.1 | 85.9 | 4.7 | 63.4 | 89.2 | 4.3 |
| TS2-Net + Ours(Large) | 51.8 | 79.1 | 7.0 | 48.6 | 79.2 | 7.4 |
| TS2-Net + Ours*(Large) | 58.3 | 82.7 | 12.0 | 60.7 | 83.1 | 9.5 |

Table 2: Performance on ActivityNet

| Method | Text2Video | | | Video2Text | | |
|---|---|---|---|---|---|---|
| | R@1 | R@5 | MnR | R@1 | R@5 | MnR |
| CLIP4ClipLuo et al. (2021) | 22.6 | 41.0 | 61.0 | 20.8 | 39.0 | 54.2 |
| CAMoECheng et al. (2021) | 22.5 | 42.6 | 56.5 | - | - | - |
| X-CLIPMa et al. (2022) | 23.3 | 43.0 | 56.0 | 22.5 | 42.2 | 50.7 |
| TS2-NetLiu et al. (2022) | 23.4 | 42.3 | 56.9 | - | - | - |
| X-PoolGorti et al. (2022) | 25.2 | 43.7 | 53.2 | - | - | - |
| CLIP-ViP Xue et al. (2022) | 25.6 | 45.3 | - | - | - | - |
| TS2-Net + Ours(Base) | 26.8 | 45.3 | 55.2 | 23.8 | 45.0 | 46.6 |
| TS2-Net + Ours*(Base) | 26.5 | 46.3 | 52.9 | 26.0 | 46.8 | 48.5 |
| CLIP-ViP + Ours(Base) | 27.0 | 46.2 | 52.9 | 25.4 | 45.2 | 46.7 |
| CLIP-ViP + Ours*(Base) | 26.9 | 46.4 | 55.3 | 27.3 | 47.0 | 45.3 |
| TS2-Net + Ours(Large) | 27.7 | 48.5 | 52.8 | 29.7 | 47.4 | 46.2 |
| TS2-Net + Ours*(Large) | 29.9 | 47.8 | 67.6 | 30.5 | 47.5 | 59.2 |

Table 3: Performance on LSMDC

| Method | Text2Video | | | Video2Text | | |
|---|---|---|---|---|---|---|
| | R@1 | R@5 | MnR | R@1 | R@5 | MnR |
| FrozenBain et al. (2021) | 34.6 | 65.0 | - | - | - | - |
| CLIP4ClipLuo et al. (2021) | 43.4 | 70.2 | 17.5 | 42.5 | 70.6 | 11.6 |
| CAMoECheng et al. (2021) | 43.8 | 71.4 | 16.3 | 45.5 | - | 10.2 |
| X-CLIPMa et al. (2022) | 45.2 | 74.0 | 14.6 | 43.1 | 72.2 | 10.9 |
| TS2-NetLiu et al. (2022) | 41.8 | 71.6 | 14.8 | - | - | - |
| HBI Jin et al. (2023) | 46.9 | 74.9 | 12.1 | 46.2 | 73.0 | 8.7 |
| CLIP-ViP Xue et al. (2022) | 48.6 | 77.1 | - | - | - | - |
| TS2-Net + Ours(Base) | 47.1 | 73.9 | 17.9 | 45.4 | 72.6 | 12.1 |
| TS2-Net + Ours*(Base) | 51.2 | 75.1 | 15.9 | 51.9 | 6.7 | 11.2 |
| CLIP-ViP + Ours(Base) | 50.9 | 78.5 | 13.2 | 44.4 | 77.1 | 11.0 |
| CLIP-ViP + Ours*(Base) | 55.5 | 80.2 | 11.0 | 58.8 | 80.0 | 8.2 |
| TS2-Net + Ours(Large) | 55.0 | 77.6 | 17.6 | 51.3 | 76.7 | 11.6 |
| TS2-Net + Ours*(Large) | 57.5 | 77.8 | 19.1 | 56.7 | 77.3 | 14.7 |

Table 4: Performance on DiDeMo

| Method | Text2Video | | | Video2Text | | |
|---|---|---|---|---|---|---|
| | R@1 | R@5 | MnR | R@1 | R@5 | MnR |
| HGRChen et al. (2020) | 35.1 | 73.5 | - | - | - | - |
| CLIPRadford et al. (2021) | 39.7 | 72.3 | 12.8 | 52.7 | 88.8 | 3.8 |
| SUPPORTPatrick et al. (2020) | 44.9 | 82.1 | 3.9 | 58.4 | 84.4 | - |
| CLIP4ClipLuo et al. (2021) | 55.9 | 89.2 | 3.9 | 73.2 | 97.1 | 1.7 |
| CLIP2VideoFang et al. (2021) | 57.3 | 90.0 | 3.6 | 76.0 | 97.7 | 1.5 |
| TS2-NetLiu et al. (2022) | 59.1 | 90.0 | 3.5 | - | - | - |
| CLIP-ViP Xue et al. (2022) | 62.3 | 90.3 | 3.4 | 77.3 | 98.1 | 1.6 |
| TS2-Net + Ours(Base) | 64.1 | 91.5 | 3.3 | 79.7 | 97.2 | 1.6 |
| TS2-Net + Ours*(Base) | 66.2 | 92.2 | 3.3 | 87.4 | 98.1 | 1.4 |
| CLIP-ViP + Ours(Base) | 64.5 | 91.3 | 3.3 | 80.1 | 98.3 | 1.5 |
| CLIP-ViP + Ours*(Base) | 67.4 | 92.5 | 3.0 | 87.6 | 98.6 | 1.4 |
| TS2-Net + Ours(Large) | 71.1 | 94.0 | 3.0 | 86.4 | 99.2 | 1.3 |
| TS2-Net + Ours*(Large) | 73.7 | 94.3 | 2.9 | 90.9 | 99.2 | 1.3 |

Table 5: Performance on VATEX

improves the T2V R@1 by 3.0% and 1.8% on TS2-Net and CLIP-ViP respectively. It is worth highlighting that despite CLIP-ViP being pretrained on a dataset encompassing over 100 million video text pairs, our proposed method still manages to further enhance its performance. Furthermore, by employing the CrossTVR Large (ViT-G/14) model and adopting the frozen CLIP model strategy, we significatn improve the T2V R@1 by 7.0% and 4.1% on TS2-Net and CLIP-ViP respectively.

**Other benchmarks.** Tables 2 to 5 provide compelling evidence of the enhanced performance achieved by the proposed CrossTVR method when combined with TS2-Net and CLIP-ViP. With ViT-B/32 vision encoder, our method improves the T2V R@1 score of TS2-Net on ActivityNet, LSMDC, DiDeMo, and VATEX by 8%, 3.4%, 5.3%, and 5.0% respectively. And our method with CLIP-ViP yields T2V R@1 scores improving by 3.9% on ActivityNet, 1.4% on LSMDC, 2.3% on DiDeMo, and 2.2% on VATEX. These outcomes consistently demonstrate that the proposed CrossTVR re-ranker effectively enhances the performance of cosine similarity methods. To further emphasize the scalability of CrossTVR to larger pre-trained models, we extend its integration to ViT-G within the framework of TS2-Net and also acheives consistently performance increase.

## 4.3 QUALITATIVE ANALYSIS

In order to assess the efficacy of our proposed CrossTVR, a qualitative analysis of the text-to-video results is conducted using multiple examples from the MSRVTT dataset, as depicted in Figure 3. The top is the retrieved text, and the last two rows are the rank1 retrieval results of TS2Net and our

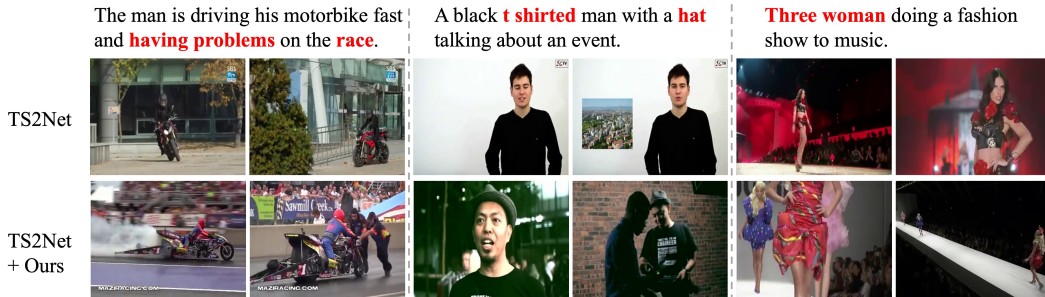

Figure 3: Visualization of text video retrieval on MSRVTT.

method. We highlight the words in the text that do not match the retrieval in red. We can see that our method retrieves fine-grained vision features based on the query text. Specifically, the initial example illustrates that the first stage TS2-Net solely captures the phrase "the man is driving his motorbike", whereas our CrossTVR further comprehends the additional context of "having problems on the race". In the second example, our CrossTVR accurately retrieves the specific object "hat" and effectively distinguishes between "shirt" and "t-shirt".

The effectiveness of the cross attention module in enhancing the fine-grained correspondences between textual and visual features at both the frame and video levels is demonstrated in Figure 4. We utilize the Grad-CAM Selvaraju et al. (2017) to visualize the cross attention map in the relevant region of the video. As depicted in Figure 4 (a), our method accurately associates the subtle movement of "pushing" with the corresponding words at the video level. Furthermore, in Figure4 (b), our approach successfully identifies the minute objects of "ball" and "pit" at the frame level.

## 4.4 ABLATION STUDY

**Effective components** In order to assess the influence of various components, we incrementally introduce different elements and analyze their impact based on TS2-Net, as presented in Table 6. Firstly, the inclusion of video level cross attention results in a noteworthy 1.4% performance boost. Subsequently, the addition of frame level cross attention in a hierarchical manner further contributes to an overall improvement of 2.6%. Moreover, by implementing parameter sharing between video text attention and frame text attention, an additional gain of 0.2% is achieved. Additionally, the adoption of a hard negative mining strategy facilitates the acquisition of intricate details by the model.Notably, the statistics reveal that the Video2Text task exhibits similar performance patterns.

**Design strategy of cross attention module** We conducted an analysis of various design strategies for the cross attention module, as outlined in Table 7. In the stage 2, given $T$ frames with $N$ tokens for each frame, we initially employed a simple token averaging approach, followed by cross attention between the text and the averaged $N$ tokens. This approach yielded a marginal improvement of 0.3% in Text2Video R@1. Subsequently, we explored the utilization of the [CLS] token from each frame, employing cross attention between the text and the $T$ [CLS] tokens. Notably, this alternative method resulted in a more substantial improvement of 1.0% in Text2Video R@1. These findings suggest that the average tokens and [CLS] tokens may not effectively capture the finer-grained and comprehensive information required to further enhance the existing cosine similarity network. Furthermore, we introduced video text attention and frame text attention as separate modules, leading

| Method | Text2Video | | | Video2Text | | |
|---|---|---|---|---|---|---|
| | R@1 | R@5 | MnR | R@1 | R@5 | MnR |
| TS2-Net | 47.0 | 74.5 | 13.0 | 45.3 | 74.1 | 9.2 |
| + Video Level | 48.4 | 74.7 | 12.1 | 46.1 | 75.4 | 8.8 |
| + Frame Level | 49.6 | 75.5 | 12.1 | 46.8 | 76.0 | 8.8 |
| + Sharing | 49.8 | 75.6 | 12.0 | 47.0 | 76.1 | 8.1 |
| + Hard | 50.0 | 75.7 | 12.0 | 47.1 | 76.6 | 8.1 |

Table 6: Ablation study of effective components of our method on MSRVTT.

| Method | Text2Video | | | Video2Text | | |
|---|---|---|---|---|---|---|
| | R@1 | R@5 | MnR | R@1 | R@5 | MnR |
| Average | 47.3 | 75.4 | 12.1 | 44.7 | 75.2 | 8.8 |
| CLS Token | 48.0 | 75.2 | 12.1 | 45.7 | 75.1 | 8.8 |
| Video Level | 48.4 | 74.7 | 12.1 | 46.1 | 75.4 | 8.8 |
| Frame Level | 48.7 | 75.0 | 12.1 | 46.1 | 75.3 | 8.8 |
| Frame Video Sum | 49.1 | 75.3 | 12.1 | 45.8 | 74.6 | 8.0 |
| CrossTVR | 50.0 | 75.7 | 12.0 | 47.1 | 76.6 | 8.1 |

Table 7: Design of cross attention mechanisms.

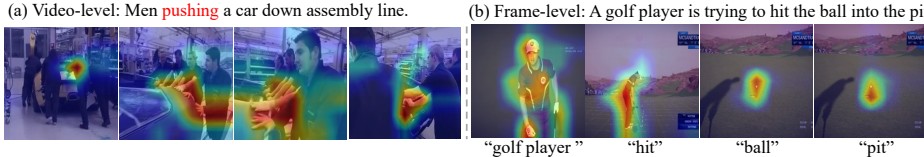

(a) Video-level: Men pushing a car down assembly line.    (b) Frame-level: A golf player is trying to hit the ball into the pit.

"golf player"    "hit"    "ball"    "pit"

Figure 4: Visualization of the cross attention map of the video text attention and frame text attention.

| Method | Text2Video | | Video2Text | | Time (s) |
|---|---|---|---|---|---|
| | R@1 | R@5 | R@1 | R@5 | |
| CLIP4Clip Luo et al. (2021) | 44.5 | 71.4 | 42.7 | 70.9 | 7.06 |
| CLIP4Clip+Ours(Base) | 47.0 | 73.8 | 44.1 | 71.9 | 7.21 |
| Xpool Gorti et al. (2022) | 46.9 | 72.8 | 44.4 | 73.3 | 12.17 |
| Xpool+Ours(Base) | 48.1 | 74.5 | 47.8 | 75.2 | 12.38 |

| Method | Vision Encoder | Text2Video R@1 | Batch Size | Memory (GB) | Training hour |
|---|---|---|---|---|---|
| CLIP4Clip | ViT-B/32 | 44.5 | 128 | 41 | 9 |
| CLIP4Clip | ViT-G/14 | 47.4 | 48 | 546 | 50 |
| Ours | ViT-B/32 | 50.0 | 128 | 41 | 18 |
| Ours | ViT-G/14 | 54.0 | 128 | 50 | 21 |

Table 8: Collaborating with different cosine similarity based methods on MSRVTT dataset. Time shows inference speed for indexing 1000 videos with one query text on one NVIDIA A100 card.

Table 9: From ViT-B to ViT-G, the increase of memory cost and training cost of our method is smaller than end-to-end finetuned CLIP4Clip. We report the accuracy on MSRVTT dataset.

to respective improvements of 1.4% and 1.7% in Text2Video R@1. To combine the outcomes of both modules, a straightforward approach involved summing the frame text matching result and the video text matching result, resulting in an additional improvement of 2.1% in T2V R@1. Additionally, by adopting a hierarchical utilization of both modules, CrossTVR achieved the highest retrieval accuracy, with a significant improvement of 3.0% in T2V R@1. These findings underscore the effectiveness of our multi-grained cross attention modules in video retrieval task.

**Collaborating with different cosine similarity based methods**   In addition to TS2Net Liu et al. (2022) and CLIP-ViP Xue et al. (2022), we have employed CrossTVR in conjunction with two other advanced cosine similarity-based methods Luo et al. (2021); Gorti et al. (2022) in Table 8. The results demonstrate the seamless integration of the proposed CrossTVR with various cosine similarity-based methods Luo et al. (2021); Liu et al. (2022); Gorti et al. (2022); Xue et al. (2022). Specifically, the findings reveal significant improvements of 2.5% for CLIP4Clip Luo et al. (2021) and 1.2% for X-pool Gorti et al. (2022) in terms of Text2Video R@1 when combined with CrossTVR. Furthermore, the last column of Table 8 shows the inference time remains virtually unchanged with and without the cross attention module. Consequently, we contend that our cross attention module holds the potential to serve as a potent performance-boosting component for the majority of cosine similarity-based TVR methods, both present and future.

**Vision encoder scalability**   Table 9 presents a comprehensive overview of the GPU memory cost and training speed associated with the end-to-end finetuning method Luo et al. (2021) and CrossTVR, employing various vision encoders. In order to provide a fair comparison, we have re-implemented the representative similarity-based approach CLIP4Clip Luo et al. (2021) using the ViT-G architecture and trained it on eight NVIDIA A100 cards with a memory capacity of 80G, utilizing the maximum batch size that can be accommodated before exhausting the GPU memory. It is worth noting that as we transition from ViT-B to ViT-G, CLIP4Clip exhibits a more than ten-fold increase in GPU memory consumption, whereas our method only experiences a modest 22% increment. Furthermore, when employing the same ViT-G architecture, our method showcases a remarkable reduction of 91% in memory utilization and a 58% decrease in training time. These findings clearly indicate that our method exhibits superior scalability to large-scale pre-trained models compared with finetuning methods thanks to the frozen vision encoder strategy.

## 5   CONCLUSION AND DISCUSSION OF BROADER IMPACT

This paper introduces CrossTVR, a novel approach for exploring comprehensive and fine-grained text-video interaction information. Experimental results demonstrate the superiority of our CrossTVR method on various TVR benchmarks, showcasing its ability to effectively handle large pre-trained vision models. Nonetheless, it is important to note that the overall performance of CrossTVR is still constrained by the performance of the retriever. Moreover, while CrossTVR offers a powerful tool for text video retrieval, it is crucial to consider the potential risks associated with the misuse of video understanding models or their outputs, such as unauthorized surveillance.

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

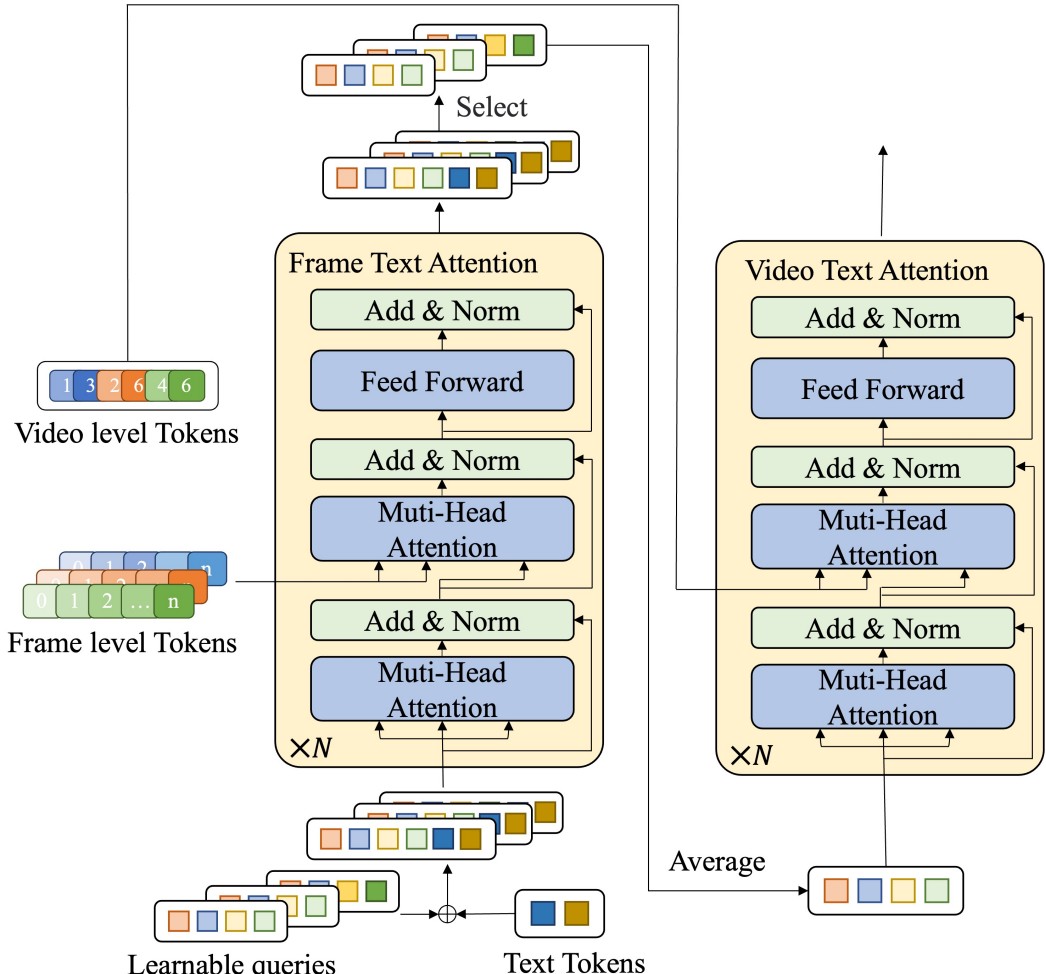

Figure 5: Details of Cross Attention Mechanism

## APPENDIX

## A    BENEFITS OF RE-RANKING

The re-rank number $K$ is a crucial aspect on retrieval performance. In order to investigate this relationship, we conduct experiments on the ViT-B network using the MSRVTT dataset. We vary the re-rank number from 2 to 19 and evaluate the results. As illustrated in Figure 6, we observe a gradual improvement in R@5 and R@10 Text2Video scores as $K$ increases. This suggests that our method is effective across different re-rank numbers. However, it is important to note that excessively large values of $K$ can lead to a decrease in retrieval efficiency. To strike a balance between efficiency and accuracy, we set $K = 15$ for all our experiments.

## B    CROSS ATTENTION MECHANISM

We present the details of cross attention mechanism in Figure 5. Specifically, for frame text attention, learnable queries are first concatenated with text tokens. These queries interact through self attention layer and then interact with each frame features in cross attention layer. These queries feed into video text attention after averaging. For video text attention, these queries interact with the selected video token in cross attention.

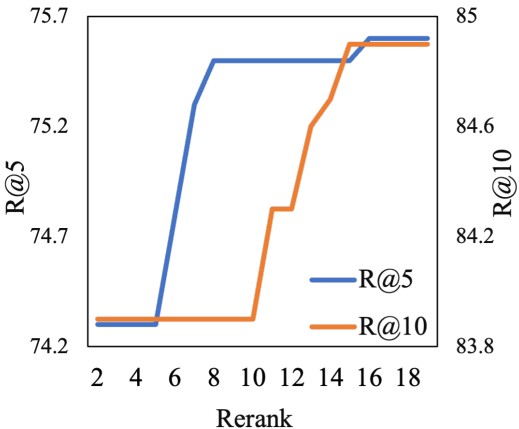

Figure 6: Different re-rank numbers.

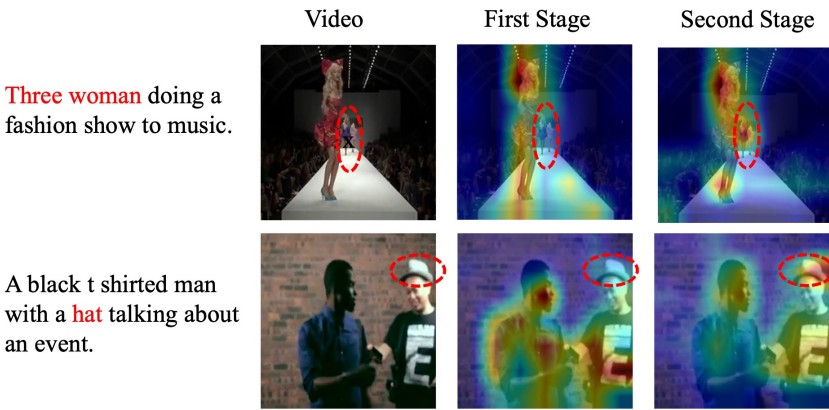

Figure 7: Fine-grained feature attention map.

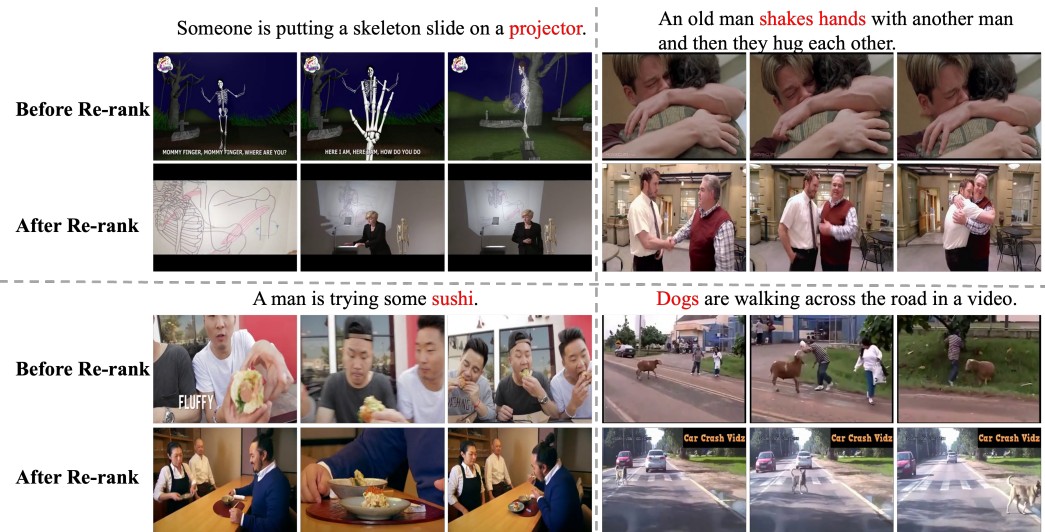

Figure 8: Text to video retrieval results.

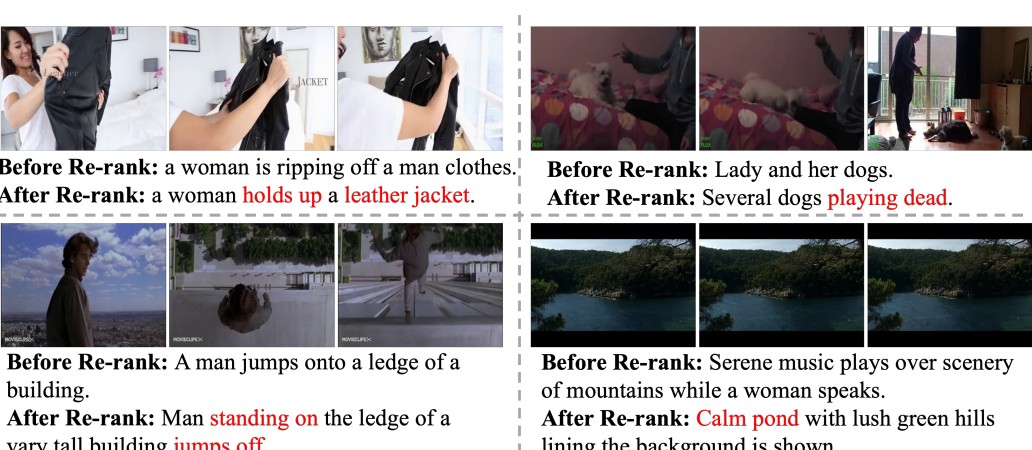

Figure 9: Video to text retrieval results.

