# OpenReview forum: "CrossTVR: Multi-Grained Re-Ranker for Text Video Retrieval with Frozen Image Encoders"
_ICLR.cc/2024/Conference — Submitted to ICLR 2024_

### Official Review · Reviewer_qf6J · 2023-10-16

**Soundness:** 3 good
**Presentation:** 3 good
**Contribution:** 2 fair
**Rating:** 6
**Confidence:** 5

**Summary:**

This paper aims at text-video retrieval in the domain of vision-and-language pretraining and understanding. Similar to general information retrieval or multimedia retrieval system, the authors proposed a two-stage or coarse-to-fine retrieval framework. In the first stage, the consine similarity network is trained with contrastive learning loss. Thus, a general text and video matching score could be obtained by such a model. The main contribution is the proposed re-ranker for fine-grained ranking in the second stage. Specifically, the authors proposed a novel cross attention module called multi-grained text-video cross attention in order to compute text-frame level attention and text-video level attention respectively. Finally, this prototype has been verified on several popular text-video retrieval benchmarks.

**Strengths:**

1. The motivation of this work is clear and is easy to follow.

2. I think the overall technology roadmap is in the right way which includes: (a) a course-to-fine retrieval framework for the sake of efficiacy and effiency trade-off; (b) freezing the image encoders to make the training process practical and affordable;

3. The proposed  multi-grained text-video cross attention mechanism is brilliant and makes sense. Specifically, spatial text attention module is proposed at the frame level in order to discover small objects or entities. And temporal text attention is designed to capture subtle movement.

**Weaknesses:**

1. It seems that the experimental results do not include retrieval performance (e.g. stage 1 / stage 2/ pre- and post-processing), which is important in terms of a cross-modal retrieval topic.

2. Again, in terms of a retrieval task, it is important to verify the effectiveness under a huge database setting. However, the most large dataset only contains 118,081 videos which is much less than industiral scales such as YouTube. At least, more distractors should be included if scaling positive text-video pairs is a concern.

3. I think the visualization is not enough. For example, the retrieval results of different work or the effect of proposed re-ranking could be included as supplementary materials if space limitation is a concern.  Besides, the visualization should verify the effectiveness of proposed saptial text attention and temporal text attention.

**Questions:**

Besides the problems in the above weaknesses part, there are a few other questions listed as follows.

[Q1] The token selector is employed as a trade-off between missing detailed information and bringing redundant information issues. The implementation of such a module is a MLP followed by a Softmax Layer, which predicts each token's importance score and selects the M most informative tokens as output. The question is what is the ground truth (GT) for such predictions? It seems that it is even impossible for human beings to label. Note that this is not a single label classfication problem.

[Q2] From the ablation study Table 6, the hard negative mining module makes little contribution to the final results, which is below my expectation. Is there any possible explanations?

[Q3] It is quite often to incorporate query expansion for the sake of increasing recall. Is it still effective after the 2nd stage re-ranking? It is not a necessary ablation study but could be considered as a option.

---

> ### Author Response · Authors · 2023-11-22
>
> ### 1. Experimental results in stage1/stage2/post-processing.
>
> The results of our searches are in Tables 1-5. The original cosine similarity method 'CLIP-VIP' is the stage1 result.  The cosine similarity method togher with our method  'CLIP-VIP + Ours' is the stage 2 result. The method denoted with (*) is the post-processing result. For better presentation, we restate  in the following table the results of the stage1/stage2/post-processing in the MSRVTT dataset.
>
> |                                 | R@1  | R@5  | MnR  |
> |---------------------------------|------|------|------|
> | CLIP-VIP (stage1)                   | 50.1 | 74.8 | -    |
> | CLIP-VIP+Ours  (stage2)         | 51.9 | 76.8 | 14.3 |
> | CLIP-VIP+Ours*(post-processing) | 56.9 | 80.0 | 11.5 |
> | TS2-Net (stage1)                      | 47.0 | 74.5 | 13.0  |
> | TS2-Net+Ours  (stage2)         | 50.0 | 75.7 | 12.0 |
> | TS2-Net+Ours*(post-processing) | 52.8 | 78.9 | 11.2 |
>
> ### 2.Larger datasets and scaling video-text pairs. More distractors should be included if scaling positive text-video pairs is a concern.
>
> Yes, we do not have such a industrial scale video retrieval dataset like YouTube.  We create a larger validation dataset by combining the vallidation set of MSRVTT, LSMDC, and Didemo dataset.  LSMDC datasets contains 118,081 videos  is currently the largest video-text retrieval public dataset for scholarly.  And MSVTT datasets contains 20 titles per video, with up to 200,000 video text pairs. Didemo contains more than 10,000 videos and 40,000 captions,
> We use the positive pairs from MSRVTT as groundtruth and use validatiaon set from LSMDC and Didemo as distractors . We use the model trained on MSRVTT dataset for evaluation. The following results show that our method is still effective in large scale retrieval dataset.
>
> |                                 | R@1  | R@5  | MnR  |
> |---------------------------------|------|------|------|
> | TS2-Net (stage1)               |  37.2   |63.5   | 44.1  |
> | TS2-Net+Ours  (stage2)         | 40.5 | 67.8   | 43.8 |
> | TS2-Net+Ours*(post-processing) | 42.7 | 68.0 | 47.0 |
>
> ### 3.More visualization results about re-ranking and spatial text attention and temporal text  attention.
>
> We add more text to video re-ranking visualization results in Figure.7 and video to text re-ranking results in Figure. 8 of the supplementary material, and visualizations of spatial attention and temporal attention are presented in Figure 5 of the paper. Spatial attention allows us to identify small objects at the frame level, while temporal attention helps us to discriminate continuous actions at the video level.
>
> ### 4. The ground truth of Token selector predicted importance score.
>
> Yes, the token selector aim to predict the most sailent video level features.  As we do not have the groundtruth of importance score, the prediction may be not accurate. However, in frame-level cross attention, we use all tokens to make sure all details information are not missed. This is the reason why combine video level cross attention and frame level cross attention can further improves the performance.
>
> ### 5.Hard negative sampling has little contribution to the final result.Is there any possible explanations?
>
> The effect of hard negative sampling method is sensitive to batch size. The hard negative sampling method usually performs better using large batch size. Prevous vision language pretraining method such as ALBEF and BLIP use a batch of 512 and 2880, while we set batch size to 128 for all of our experiments due to limited training resources. We also tried a smaller batch size of 64 on MSRVTT dataset and get a marginal performance contribution.
>
> ### 6.Is query expansion effective in the 2nd stage re-ranking?
>
> We have not yet experimented on query expansion for fair comparison since other methods do not use it . We followed previous work using DSL post-processing, which serves as a reviser to correct the similarity matrix and achieves the dual optimal match. We use (*) to denote methods use DSL in Table 1~5. It shows the DSL is equally effective in stage one and stage two.

---

> > ### Comment · Reviewer_qf6J · 2023-11-23
> >
> > Thanks for your furhter response. When referring to retrival performance, I particularly mean the processing time in each stage (stage 1 / stage 2/ pre- and post-processing), which is vital for retrieval related tasks.

---

> ### Author Response · Authors · 2023-11-23
>
> We measure the stage1 and stage2 time performance in Table 8 of the main paper. For better presentation, we restate in the following table the processing time  to index 1000 videos with one query text. In the following table, the stage1 time is the sum of time to encode 1 text, the time to encode1000 videos, and the time to compute cosine similarity. The stage1+stage2 time is the sum of the stage1 time and the time to compute the cross attention. We can see stage2 only increase 2\% retreival time. Since we use dual softmax  as the post procssing step, which only involves  matrix calcuation, the post-procssing time is very small and can be ignored comparing to retrieval time.
>
> | Method                         | Times(s) |
> |--------------------------------|----------|
> | CLIP4CLIP (stage1)             | 7.06     |
> | CLIP4CLIP+ours (stage1+stage2) | 7.21     |
> | CLIP4CLIP+ours (stage1+stage2+post procssing) | 7.21     |
> | Xpool (stage1)                 | 12.17    |
> | Xpoll+ours (stage1+stage2)     | 12.38    |
> | Xpoll+ours (stage1+stage2+post procssing)     | 12.38    |

---

> > ### Comment · Reviewer_qf6J · 2023-12-01
> >
> > Thanks for you further response. It seems that the overall retrieval time is far from a real-time system although the stage 2 only increases very little time cost. I understand that your main contribtuions focus on the deep model part.

---

### Official Review · Reviewer_TCyW · 2023-10-31

**Soundness:** 3 good
**Presentation:** 2 fair
**Contribution:** 3 good
**Rating:** 5
**Confidence:** 5

**Summary:**

This paper puts forward a new method for text-video retrieval. The core contribution of the study revolves around a multi-grained re-ranker that is designed to effectively retrieve relevant videos using text as a query. The paper also proposes a design strategy for the cross attention module by collaborating with different cosine similarity based methods, which further enhances the retrieval efficiency. And the experiments show good performance.

**Strengths:**

1. The paper is articulated with clarity and precision.
2. The multi-grained re-ranker is an innovative approach that can potentially enhance text-video retrieval's efficacy. Furthermore, the use of frozen image encoders in the context of text-video retrieval is a commendable attempt at exploring uncharted territories.

**Weaknesses:**

1. Explain the specific implementation and mechanism of video-level and frame-level cross attention. And how do these components operate and interact.
2. Although frozen vision encoder reduce computational costs, this approach may limit the model's flexibility and ability to adapt to different tasks.
3. There are some formatting errors, i.e Figure 4.

**Questions:**

1. Please elaborate on how to match and align between video and text.
2. In addition to TS2Net, the results of other existing methods are given to better evaluate the performance improvement and advantages of this method

---

> ### Author Response · Authors · 2023-11-22
>
> ### 1. Explain the specific implementation and mechanism of video-level and frame-level cross attention.
>
> Please refer to the supplementary material Fig. 5, which shows the details of our video-level and frame-level cross attention.  Specifically, for frame text attention, learnable queries are first concatenated with text tokens. These queries interact through self attention layer and then interact with each frame features in cross attention layer. These queries feed into video text attention after averaging. For video text attention, these queries interact with the selected video token in cross attention.
>
> ### 2. Although frozen vision encoder reduce computational costs, this approach may limit the model's flexibility and ability to adapt to different tasks.
>
> Yes, we have not try other video understanding tasks like video QA, video caption so we do not know whether the frozen vison encoder strategy is also applicable on these tasks. However,  we are certain that finetuning the whole vision encoder will not perform worse than the frozen vision encoder. Therefore, when adapting our method to other tasks, we can finetune the vision encoder to increase the performance.
>
> ### 3. There are some formatting errors, i.e Figure 4.
>
> We have revised Figure 4 and update the manuscript.
>
> ### 4. Please elaborate on how to match and align between video and text.
>
> We use contrastive loss and video text matching loss to supervise the alignment between video and text in stage 1 and stage 2, respectively.  The contrastive loss compare each text to all videos in a batch. And the matching loss compute the text to the hardest video case in a batch with the help of hard negative mining strategy.  Fig. 5 in  supplementary material shows the details of the interaction between text and video in cross attention.
>
> ### 5. In addition to TS2Net, the results of other existing methods are given to better evaluate the performance improvement and advantages of this method.
>
> We report the results our method on top of four SOTA cosine similarity methods in Tables 1 and 8 of paper. Specifically on the t2v R@1 metric for the MSRVTT dataset, our method improves by 2.5, 1.2, 3.0, and 1.8 on the Clip4Clip, Xpool, TS2-Net, and clip-vip methods, respectively.

---

### Official Review · Reviewer_dEsS · 2023-11-01

**Soundness:** 2 fair
**Presentation:** 2 fair
**Contribution:** 2 fair
**Rating:** 5
**Confidence:** 4

**Summary:**

This paper proposes a multi-grained re-ranker for text video retrieval with frozen image encoders, named CrossTVR. Although experimental results have demonstrated the effectiveness of this paper, the main contributions of this paper are not clear enough.

**Strengths:**

1.The authors develop a novel method to capture comprehensive and fine-grained text-video interaction cues, which uses a multi-grained video text cross attention transformer module to enhance the retrieval process.
2.The corresponding experiments validate the effectiveness of the proposed method on five public text-video retrieval benchmarks and show a significant improvement in solution acceptance rates.

**Weaknesses:**

1.	The Abstract is not well written and it is not easy to understand the contributions. Recommended to rewrite and well present your contributions.
2.	For the first stage, this paper uses an existing cosine similarity network, the novelty is limited for me.
3.	For the technique part, the motivation for multi-grained video text cross-attention is unclear. How to enhance the fine-grained correspondences by the cross-attention module.
4.	Furthermore, what specific attention mechanism is being employed? Please explain this work's fundamental research insight.
5.	The main contributions of this paper are not clear enough to show the advantages and the novelty is not well explained. Please summary the main contributions in the introduction section.
6.	The organization of this manuscript is a mess, especially the order of the discussion of Tables 1 to 5. The related discussion should be even clearer.
7.	In addition, the authors should revise the manuscript thoroughly. Here are some data errors and typos as listed below (including but not limited to). The authors need to make the necessary changes.
1)	In Table 5, there are data errors that need to be corrected.
2)	In the 4.2 section, ‘moreover, our method, which leverages CLIP-VIP, achieves …’

**Questions:**

None

---

> ### Author Response · Authors · 2023-11-22
>
> ### 1. Rewrite Abstract.
>
> State-of-the-art text-video retrieval (TVR) methods typically utilize CLIP and cosine similarity for efficient retrieval.
> Meanwhile, cross attention methods, which employ a transformer decoder to compute attention between each text query and all frames in a video, offer a more comprehensive interaction between text and videos.
> However, these methods lack important fine-grained information as they directly compute attention between text and video-level tokens.
> To address this issue, we propose a fine grained re-ranker named CrossTVR with a multi-grained video text cross attention module to capture fine-grained multimodal information from both frame and video level. The re-ranker only improves the top K similar results from the cosine similarity network, so the inference can be performed efficiently.
> To reduce the additional training cost by the multi-grained cross attention module, we freeze the vision backbone and only train the video text cross attention matching header, enabling scalability to larger pre-trained vision models like ViT-G, resulting in improved retrieval performance.
> Experiments on text video retrieval datasets demonstrate the effectiveness and scalability of our proposed CrossTVR compared to state-of-the-art approaches.
>
> ### 2. For the first stage, this paper uses an existing cosine similarity network, the novelty is limited .
>
> Yes, we clarify Stage 1 is not our contribution. CrossTVR is a  re-ranker which is compatible with existing retrieval methods.
>
> ### 3. The motivation for multi-grained video text cross-attention is unclear. How to enhance the fine-grained correspondences by the cross-attention module.
>
> In previous work on video text retrieval, one approach is to use the embeddings of video and text for cosine similarity, which is efficient but limited in accuracy. While another is to use all video tokens and text for cross attention, which costs expensive computation. We consider an method that minimizes computational effort while maximizing the retention of frame-level and video-level information to achieve full interaction between video and text in cross attention.
> In the first stage, existing cosine similarity network can retrives Top K most similar results with high recall. Then in the second stage, the goal of re-ranker is to find the minor differents between these K results, which we believe fine grained correspondences are critical. Therefore, we use frame text attention to achieve the interaction between frame and text to obtain fine-grained features. And we further visualize the fine-grained feature attention map to show our fine-grained feature extraction capability in supplymentary meterial Fig. 7.
>
> ### 4. What specific attention mechanism is being employed? Please explain this work's fundamental research insight.
>
> Our attention mechanism is built on top of BERT, which is also widely used in previous multimodal work such as ALBEF, BLIP, X-VLM. We show the details of our cross attention in the supplementary material Fig. 5. The attention mechanism is not novel, the way to fuse frame level information and video level information is critical in our method. Specifically, We use a set of  fixed number learnable queries to sequentially  interact with text, frame level , and video level information, which extracts relavent information from each other.
>
> ### 5. The main contributions of this paper are not clear enough to show the advantages and the novelty is not well explained. Please summary the main contributions in the introduction section.
>
> In summary, The main contributions of this work are as follows:
>
> * We propose a multi-grained re-ranker called CrossTVR, which achieves comprehensive interaction between text and video at the frame level and video level.
> * As a re-ranker, our method can be widely applied to existing cosine similarity-based methods and effectively improve the SOTA retrieval performance with marginal additional computation cost.
> * Benefiting from our freezing visual coder training method, our approach can scale to larger pretrain visual models with small computational resources.
>
> ### 6. There are a few minor issues in the writing of this paper.
>
> We follow the advice and have fixed the writing issues in the uploaded revised manuscript.

---

### Meta-Review · Area_Chair_Sbm4 · 2023-12-05

**Metareview:**

The paper uses a two-stage retrieval framework for text-to-video retrieval and proposes a multi-grained reranker in the second stage. The reranker uses cross-attention from text to both temporal and spatial video features. Experiments on several text-video retrieval datasets demonstrate the effectiveness of the proposed approach.

The major concern from the reviewers is that the paper is not well written with unclear contribution and method descriptions. The AC agrees with the majority of reviewers and recommends rejection.

**Justification For Why Not Higher Score:**

Explained in the metareview.

**Justification For Why Not Lower Score:**

N/A

---

### Decision · Program_Chairs · 2024-01-16

Reject